# Short-Term Supplemental Dietary Potassium from Potato and Potassium Gluconate: Effect on Calcium Retention and Urinary pH in Pre-Hypertensive-to-Hypertensive Adults

**DOI:** 10.3390/nu13124399

**Published:** 2021-12-09

**Authors:** Michael S. Stone, Berdine R. Martin, Connie M. Weaver

**Affiliations:** Department of Nutrition Science, Purdue University, West Lafayette, IN 47907, USA; Stonemi13@aol.com (M.S.S.); berdinemartin44@gmail.com (B.R.M.)

**Keywords:** potassium, calcium, calcium retention, acid-ash hypothesis

## Abstract

Potassium supplementation has been associated with reduced urinary calcium (Ca) excretion and increased Ca balance. Dietary interventions assessing the impact of potassium on bone are lacking. In this secondary analysis of a study designed primarily to determine blood pressure effects, we assessed the effects of potassium intake from potato sources and a potassium supplement on urinary Ca, urine pH, and Ca balance. Thirty men (*n* = 15) and women (*n* = 15) with a mean ± SD age and BMI of 48.2 ± 15 years and 31.4 ± 6.1 kg/m^2^, respectively, were enrolled in a cross-over, randomized control feeding trial. Participants were assigned to a random order of four 16-day dietary potassium interventions including a basal diet (control) of 2300 mg/day (~60 mmol/day) of potassium, and three phases of an additional 1000 mg/day (3300 mg/day(~85 mmol/day) total) of potassium in the form of potatoes (baked, boiled, or pan-heated), French fries (FF), or a potassium (K)-gluconate supplement. Calcium intake for all diets was approximately 700–800 mg/day. Using a mixed model ANOVA there was a significantly lower urinary Ca excretion in the K-gluconate phase (96 ± 10 mg/day) compared to the control (115 ± 10 mg/day; *p* = 0.027) and potato (114 ± 10 mg/day; *p* = 0.033). In addition, there was a significant difference in urinary pH between the supplement and control phases (6.54 ± 0.16 vs. 6.08 ± 0.18; *p* = 0.0036). There were no significant differences in Ca retention. An increased potassium intake via K-gluconate supplementation may favorably influence urinary Ca excretion and urine pH. This trial was registered at ClinicalTrials.gov as NCT02697708.

## 1. Introduction

Osteoporosis is a global health problem characterized by a severe reduction in bone mass leading to increased fracture risk [1]. More than 200 million people worldwide suffer from osteoporosis, and it is estimated that 40% of postmenopausal women and 30% of men will have an osteoporotic fracture in their lifetime [2,3]. Peak bone mass is achieved by the third decade of life, after which bone loss accelerates with aging in both men and women. Bone maintenance optimization is determined by numerous influences that affect turnover throughout the lifespan, many of which are modifiable lifestyle factors [4]. 

Adequate potassium intake may benefit bone through its effect on acid-base balance [5,6]. Western diets high in meats and cereal grains produce an environment of low-grade metabolic acidosis and buffering of this increased acid load may rely on alkaline calcium (Ca) salts derived from the dissolution of bone tissue [4]. Potassium salts, produced from the consumption and metabolism of fruits and vegetables or alkalizing supplements (e.g., potassium supplements), may also yield alkaline precursors that help maintain pH homeostasis (~7.4). Excess systemic acid may impact bone via two mechanisms: (1) pH buffered through the dissolution of the bone matrix, and (2) cell-based mechanisms (e.g., up-regulation of bone resorbing cells (osteoclasts)) [4,5,6]. Despite physiological plausibility, the acid-base balance theory has been met with empirical challenges [7,8]. Yet, there are few other theories that define potential mechanisms of potassium benefit to bone [9].

Cross-sectional, observational studies show a consistent bone benefit with increased dietary potassium associated with higher fruit and vegetable intakes, especially in adults and older men and women [10,11,12,13,14,15]. Dietary interventions assessing the impact of potassium on bone are lacking, although in an ancillary study to the Dietary Approaches to Stop Hypertension (DASH) trial, osteocalcin and C-telopeptide (CTX) were both significantly reduced from baseline by the DASH dietary pattern compared to control [16]. Evidence from clinical supplementation trials suggests that potassium intake decreases Ca excretion [17,18,19], and may increase overall Ca balance [20,21], although findings have been inconsistent [22]. Studies have also examined the relationship between potassium intake and bone turnover markers, showing decreases in resorption markers of C- and N-telopeptide and increases in procollagen type I N-terminal propeptide, with potassium supplementation, along with decreases in urinary Ca [20,23,24]. The few randomized control trials that assessed the effect of potassium supplementation on bone mineral density (BMD) had mixed results [13,25,26], and none assessed effects on osteoporotic fracture risk. Overall, the potential bone benefit of potassium intake has primarily been seen with potassium supplementation at high doses (2300–3500 mg/day (60–90 mmol/day)) [9]. It is yet to be determined if increases in dietary potassium can produce the same effect, and if so, how this effect is being mediated.

Potato sources make up approximately 20% of vegetable intake in the American diet; for overall potassium intake, white potatoes and French fries represent ~7 and 3%, respectively [27,28]. Thus, potatoes are an excellent potassium food source to study.

The purpose of this paper is to report findings of the secondary outcomes of a randomized, controlled feeding trial designed primarily to examine the effect of increased dietary potassium from potato sources and a potassium-gluconate supplement compared to a control diet on blood pressure. The primary outcomes have been previously reported [29]. The research in this report aimed to assess the effect of increased dietary potassium from potato sources and a potassium-gluconate supplement on the Ca economy compared to a control diet. It was hypothesized that the interventions with an increased potassium intake would decrease Ca urinary excretion and increase calcium retention and urinary pH compared to the control.

## 2. Materials and Methods

### 2.1. Subjects

Study procedures have been described elsewhere [29]. Briefly, subjects were pre-hypertensive-to-hypertensive (systolic blood pressure >120 mmHg) men and women (*n* = 30, age 21 y and older). Exclusion criteria included: cardiovascular or renal disease, gastral intestinal disorder/disease, liver disease, cancer, pancreatitis, pregnancy/lactation, smoking, excessive alcohol intake or illegal drug use, intolerance to study foods, use of dietary supplements during study periods, and excessive weight loss (>3 kg in last 2 months). Those using medications that affected electrolyte balance and/or to treat hyper or hypotension were also excluded. Hypertensive monotherapy with strict adherence was allowed later in recruitment. Data were collected from May 2016 to November 2018. Study procedures were voluntary; all participants provided informed consent. This study was approved by the Purdue University Institutional Review Board (Protocol #1511016780) and is registered at ClinicalTrials.gov as NCT02697708.

### 2.2. Dietary Intervention

The parent study was a randomized, controlled feeding trial with cross-over design and primary outcomes of potassium balance and blood pressure (reported elsewhere [29]). Secondary measures assessed in this report included Ca excretion and Ca balance. After initial screening, eligible subjects entered the 4-day run-in, each day designed to represent a different study phase (control, potato, French fries, K-gluconate supplement) and to assess study adherence. After this trial period, participants were assigned to a random order (24 possible sequences) of four 16-day dietary potassium interventions including a control diet of 2300 mg/d (~60 mmol/d), and three phases of an additional 1000 mg/d (3300 mg/d(~85 mmol/d) total) of potassium in the form of potatoes (baked, boiled; no additional fat), French fries, or a K-gluconate supplement. Potassium gluconate was chosen due to its organic anion and measured bioavailability in our previous study [30]. Study phases were separated by a washout period (≥2 weeks). The menu cycle consisted of 4-days at three calorie levels (1800, 2200, and 2600 kcal/day) with the primary goal of potassium intake manipulation, while keeping other nutrients constant. Each menu was designed to include approximately 800 mg/day of Ca. Subjects picked up study foods approximately every other day, recorded their intake by using check-off sheets and were instructed to indicate the consumption of non-study foods and/or whether any foods that were not eaten. Any uneaten items were instructed to be returned and analyzed for mineral content. All meals were prepared with deionized water and weighed to the nearest one tenth gram, and deionized bottled water was also given for consumption to control water intake. Duplicates of menu cycle days were homogenized and analyzed for mineral content at Purdue and potassium supplements (K-gluconate) were analyzed by a food chemistry lab (Eurofins Food Chemistry Testing US, Inc., San Antonio, TX, USA).

### 2.3. Calcium Balance and Net Absorption

General procedures for measurement of mineral excretion and retention have been described previously [29]. Twenty-four-hour urine and feces were collected in acid washed containers. Instructions and supplies (e.g., containers for excreta) were provided. Urine was analyzed for creatinine (for compliance) and minerals (potassium, sodium, Ca). Stools were analyzed for polyethylene glycol (PEG; for sample recovery completeness) and minerals. Creatinine was assessed by a kinetic modification of Jaffe’s colorimetric assay (Cobas Mira Plus; Roche Diagnostic Systems, Nutley, NJ, USA). Urinary pH was assessed on 24-h samples collected on days 14 and 15 of each intervention using a benchtop pH meter (OAKTON Instruments, Vernon Hills, IL USA). Urine was acidified with 1% (by vol) HCl for mineral analysis. Stool samples were homogenized in ultra-pure water and HCl using a laboratory stomacher (Tekmar Co., Cincinnati, OH, USA). PEG was administered as two 500 mg capsules, instructed to be taken three times per day with each meal. Completeness of fecal collections was determined by turbidimetric assay. Fecal samples were placed in a drying oven at 50 °C for a minimum of 24 h, and then ashed in a muffle furnace at 600 °C for 96 h. Ashed samples were diluted in 1 N HNO_3_ for total mineral analysis. Mineral concentrations were measured by ICP-OES (5100 PC; Perkin Elmer, Waltham, MA, USA). All excreta samples were stored at −20 °C until analysis.

After a Ca intake equilibration period of the first 7 days and exclusion of day 16 (onsite clinical testing day), an 8-day period was assessed for Ca excretion and Ca retention for each study phase. Retention and percent (%) absorption were determined for the 8-day average using the following equations:Daily calcium balance (mg/d) = daily calcium intake (mg/d) − daily calcium excretion (mg/d) (urine and stools).
Percent (%) net absorption = daily calcium intake (mg/d) − daily calcium fecal excretion (mg/d) × 100.

### 2.4. Statistical Methods

Differences among interventions for Ca urinary loss, fecal loss, urine pH, and calcium retention (as % of intake), were analyzed using a mixed model ANOVA and Tukey post hoc adjustment for pairwise comparisons among diet interventions. Absolute Ca retention (mg/d) was analyzed using the same model with a contrast analysis for post hoc tests to examine mean differences between the control and K-gluconate phases. All statistics were performed using JMP (SAS Institute, Cary, NC, USA) software, and α was set at 0.05. We used a previous dose-response potassium supplementation study to estimate *a priori* power [21], estimating a sample size of 15 per group was needed to have 80% power with a 2-sided α of 0.05 to see a difference of 78 mg/d in Ca retention; however, Post-hoc power analyses revealed a sample size of 20 cross-over participants provided 80% power with two-sided α = 0.05 to detect differences in Ca retention of approximately 165 mg/d or 21% of daily intake, respectively. All values are reported as mean ± SE unless otherwise stated.

## 3. Results

### 3.1. Baseline Characteristics of Study Subjects

Overall, data from 30 subjects were included in the final analysis of urinary pH (Table 1), 28 subjects were included for urinary Ca excretion, and 20 subjects for whom full balance measures were available were included in the fecal Ca excretion, and Ca retention analysis. On average, subjects were middle-aged to older adults, had a BMI bordering overweight to obese, and a systolic blood pressure and diastolic blood pressure of 133.6 ± 12.2 mmHg and 85.5 ± 8.6 mmHg, respectively.

### 3.2. Subject Adherence

A total of 25 subjects completed all four interventions, one subject completed three interventions, and four subjects completed two interventions. Supplement intake compliance by pill count was 91%. For Ca fecal excretion, retention, and % absorption, all 20 subjects for which these measures are available completed all four phases of the study (Figure 1).

### 3.3. Chemical Analysis of Controlled Diets

There was some unexpected variation in potassium, sodium, and Ca content of the diets. Potassium content (mean ± SE) for the intervention phases of control, potato, French Fry (FF), and supplement were 2238 ± 44, 3008 ± 4, 2977 ± 29, and 3299 ± 43 mg/day, respectively. The potassium levels for the control and supplement phases were close to the target of 2300 and 3300 mg/day, respectively, while the potato and FF were slightly lower at approximately 3000 mg/day compared to the 3300 mg/day target. For sodium, the target for all four interventions was set at 3300 mg/day. However chemical analysis showed sodium content to be slightly higher for the potato (3417 ± 77 mg/day) and FF (3466 ± 112 mg/day), and slightly lower for the control and supplement phases (2974 ± 61 mg/day). For Ca, the target for all four interventions was set at 800 mg/day, however chemical analysis showed Ca content to be slightly lower for the control and supplement diets (765 ± 29 mg/day), as well as the FF phase (684 ± 31 mg/day). The potato diet did achieve approximately 800 mg/day of Ca (799 ± 21 mg/day).

### 3.4. Urinary Calcium Excretion and pH

There were significant differences among groups in urinary Ca excretion (*p* = 0.0007), the K-gluconate supplement (96 ± 10 mg/day) was significantly lower compared to the control (115 ± 10 mg/day; *p* = 0.027) and potato (114 ± 10 mg/day; *p* = 0.033). There were also significant differences between interventions in urinary pH (control: 6.08 ± 0.2, potato: 6.36 ± 0.1, French fries: 6.33 ± 0.1, supplement: 6.54 ± 0.2; *p* = 0.008), with the K-gluconate intervention resulting in a higher urinary pH compared to control (6.54 ± 0.2 vs. 6.08 ± 0.2; *p* = 0.0036) (Figure 2).

### 3.5. Calcium Fecal Excretion, Balance, and % Absorption

There were no significant differences (overall *p* = 0.12) in fecal Ca excretion with losses of 676 ± 72, 609 ± 67, 537 ± 60, 602 ± 54 mg/day, for control, potato, FF, and supplement interventions, respectively. However, there was a trend between the potato group and control (*p* = 0.089) by contrast analysis. Differences in Ca retention were assessed as absolute and % of intake. There were no significant differences among groups for either absolute Ca retention (all mg/day; control: 162 ± 58, potato: 162 ± 55, FF: 186 ± 57, supplement: 260 ± 46; *p* = 0.27) or retention based on percent of intake (control: 17.5 ± 7.3%, potatoes: 19.4 ± 6.8%, French fries: 21.0 ± 7.9%, supplement: 29.3 ± 5.5%; *p* = 0.41; Figure 3). Appendix A depicts absolute Ca retention data distribution for each intervention. There were no significant differences among groups for % Ca absorption (control: 31.1 ± 6.3%, potato: 31.5 ± 6.2%, French fries: 31.8 ± 7.1%, supplement: 36.7 ± 4.7%; *p* = 0.85).

## 4. Discussion

In this highly controlled feeding trial, we assessed the effects of dietary potassium from potato sources (bake or boiled and French fry) or a potassium supplement (K-gluconate) on outcomes of Ca balance, including urinary and fecal Ca excretion, Ca retention, and net percent Ca absorption. We also assessed urinary pH which has been cited as a surrogate for net acid excretion (NAE) [31]. We found significant differences among groups in urinary Ca excretion (lower in Supp vs. Potato and Control) and urine pH (higher in Supp vs. Control); however, this did not translate into differences in Ca retention, likely due to the variance added with the fecal component of balance. There were also no significant differences in % absorption among groups. Our findings are somewhat consistent with the previous literature assessing theories related to the acid-ash hypothesis; however, our results do not confirm any improvement in Ca retention, or bone benefit, based on these measures as surrogates for bone health.

The acid-ash hypothesis proposes that a diet higher in meats and cereal grains, precursors to acid metabolites (e.g., phosphates, sulfates), causes chronic acidemia over time. This can lead to the dissolution of bone tissue and the release of Ca for its alkalizing effects [32,33], causing increased urinary Ca and decreased Ca retention [34,35]. In contrast, the consumption of fruits and vegetables, and other base producing (e.g., potassium, Ca) foods or supplements (e.g., bicarbonate) would decrease urinary Ca excretion and have a protective effect on bone [32,34,36]. A recent meta-analysis shows a linear relationship between urinary net acid excretion (NAE) and urinary Ca, with the increased Ca loss related to the modern acidogenic diet amounting to an estimated 66 mg/day (1.6 mmol/day) [37]. Despite this significant finding, in a follow-up meta-analysis, researchers from the same group found no relationship between NAE and Ca balance [34]. However, this review included only five studies, all primarily assessing increased protein load, ignoring other diet factors such as fruit and vegetables or bicarbonate salts. The connection between increased (or decreased) dietary acid directly affecting bone is still unclear.

Observational studies show a consistent bone benefit with increased dietary potassium associated with higher fruit and vegetable intakes across gender and life stage [10,11,12,13,14,15,38,39]. While clinical interventions assessing the effect of increased dietary potassium on Ca balance are lacking, evidence from clinical supplementation trials suggest that potassium intake decreases urinary Ca excretion [15,16,17], and may increase Ca retention [18,19]. In postmenopausal women (*n* = 18) potassium (K)-bicarbonate (60–120 mmol/day) taken for 18 days decreased urinary Ca excretion by 76 mg/day, and increased Ca retention by 56 mg/day [20]. In a more recent randomized double-blind placebo-controlled study, Moseley and colleagues assessed the effect of potassium supplementation on Ca retention in older men and women (*n* = 52; >55 y). Subjects who were randomly assigned to six months of 60 or 90 mmol/day of potassium (K)-citrate supplementation had decreases in urinary Ca and NAE, the highest dose (90 mmol/day) resulting in positive Ca balance compared to control (0 mmol/day) [21]. The middle dose of 60 mmol K/day on top of the control diet which contained 75 mmol K/day resulted in a difference in Ca retention of approximately 110 mg/day compared to the control (*p* = 0.18). This is similar to the difference we observed (trend by contrast analysis) between our potassium supplemented periods compared to the control periods. Our crossover design was stronger than their parallel arm design, but the difference in potassium levels was one-fourth that of the Moseley study. As in the Moseley study, we found no effect of potassium supplementation on Ca absorption. While these findings seem consistent, they are few, and overall research looking at the effects of increased potassium intake (via supplements or food) on Ca balance as a surrogate for bone health is lacking.

We found significant differences in urinary Ca between the supplement and potato interventions, which was unexpected. This may be partially attributed to discrepancies in calculated and chemically analyzed study diets; actual potassium content of the potato diet was ~300 mg/d lower than supplement. This difference in planned vs. actual/analyzed intake is similar to findings of other controlled feeding trials [40,41], and could have been due to variations in potassium content of fruits and vegetables, which may differ based on season, location, and crop. Sodium content was slightly higher for the potato (3417 ± 83 mg/day) and FF (3466 ± 117 mg/day), and lower for the control and supplement phases (2974 ± 61 mg/day) than planned calculations (~3300 mg/day). High sodium intakes have been shown to increase urinary Ca losses, with a loss of approximately 24–40 mg of Ca for a Na intake of ≈2.3 g [42]. This, along with the potassium difference, may largely explain our results.

The recently published 2019 Dietary Reference Intake recommendations for potassium and sodium, assessed numerous chronic diseases in the context of adequate or increased potassium intake and risk reduction [43]. Bone health, or osteoporosis, was highlighted in this report while it has often been overlooked by others [44]. The committee focused their review on outcomes related to fractures and BMD, citing the fact that surrogate measures of bone health (e.g., urinary Ca excretion, retention, and bone turnover markers) may suggest a potential effect on bone integrity, but their interdependence is not clear in the literature. There are no randomized control trials looking at potassium intake and fracture risk, and the few trials examining potassium and BMD show both improvement [25,45], and null results [13,26].

Since the majority of the trials looking at the bone outcomes primarily manipulate potassium intake via supplementation, the committee also raised the question of potential effects being attributed to the non-potassium portions of the supplements. Jehle and colleagues found that postmenopausal women taking K-citrate showed significant increases in spine BMD after 12 months (0.89%; *p* < 0.05), whereas women in a potassium (K)-chloride (Cl) group had decreased spine BMD of (−0.98%; *p* < 0.05) [45]. Both supplements had the same potassium content (30 mmol/day). In contrast to this evidence, in a short-term supplementation study in healthy men and women urinary Ca excretions decreased after 4-days of K-chloride or K-bicarbonate, which was not seen with identical supplementation of Na-chloride or Na-bicarbonate [18]. In addition, researchers using a rat model proposed that the bone benefit from a vegetable rich diet may be related to bioactive compounds (e.g., flavonoids) rather than base excess after showing the addition of K-citrate at levels that neutralized NAE from an acidogenic diet had no effect on bone turnover [7]. These findings suggest that the form of supplementation, or the other constituents in foods are important factors in understanding the role of potassium in the acid-ash hypothesis, and whether or not potassium intake has any independent benefit to bone.

Strengths of our study include the highly controlled diet, the use of a crossover design, and comparing food and supplement forms of potassium. The comparison of forms allowed insights into potential roles of the effect of bioactives or other constituents in potatoes. Duration of the study was adequate to achieve steady state calcium retention [46]. A limitation of this study is that only one, relatively moderate Ca intake was included. A dose response design would have been stronger. It is possible that effects of potassium intake on Ca balance may have differing effects based on whether Ca intake is low, adequate, or high. As with all balance studies, there are potential sources of error with complete collection in only one direction, i.e., incomplete collection of urine and feces. Compliance with the diet was based on an honor system except for meals served in the clinic. Calcium absorption decreases with age [47], which may have affected intersubject variation, but not intrasubject variation in this crossover design. Our findings agree in part with the previous literature with Ca retention trending higher for the K-gluconate intervention vs. the control. However, we were most likely underpowered (~*n* = 50, based on post-hoc analysis). Even so, the mean Ca retention between control and potato groups was identical showing increased power would not show a benefit of food potassium on Ca retention.

Future research will first need to establish a specific role for potassium effecting Ca balance and bone health, independent of other supplement or dietary constituents that may have alkalizing, or other physiological, effects on their own. Dose-response evidence for potassium, especially potassium from foods, and how this influences Ca economy, as well as bone, also need to be examined. Here we confirm previously seen benefits on Ca excretion and NAE with a relatively low dose of additional supplemental potassium (~25 mmol/day), resulting in an overall intake of 3300 mg/day, a practical modification to the everyday diet. Overall, research to date has established a vague relationship between potassium intake and bone benefit. Whether or not this association is exclusive to the context of the acid-ash hypothesis, or if increased potassium intake can independently improve Ca economy and bone health is still not well understood.

## Figures and Tables

**Figure 1 nutrients-13-04399-f001:**
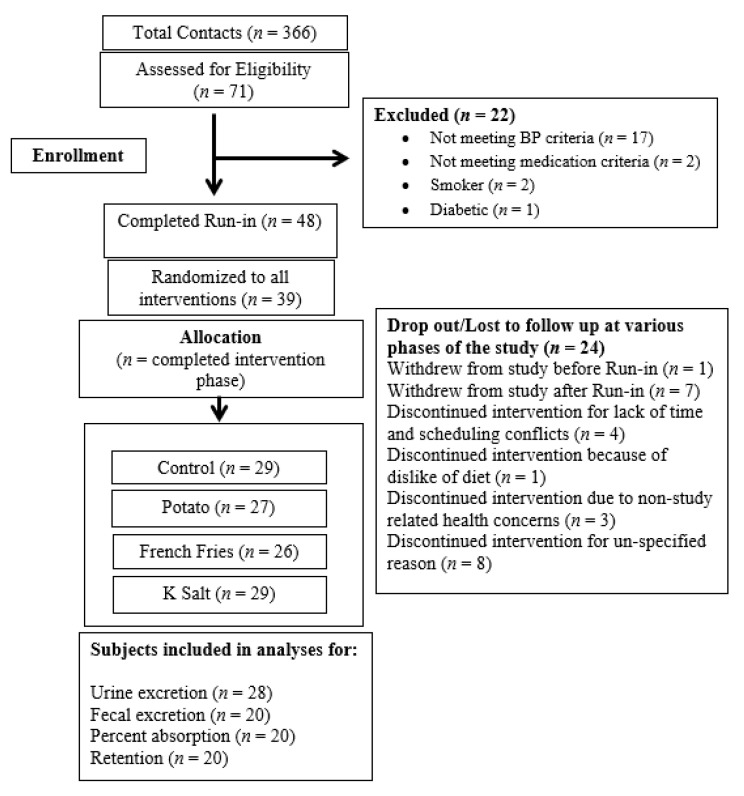
Flow Diagram of Study.

**Figure 2 nutrients-13-04399-f002:**
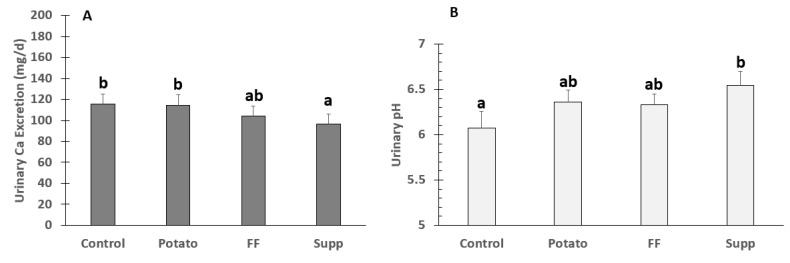
Means ± SE of (**A**) urinary Ca Excretion (*n* = 28) and (**B**) pH (*n* = 30) in adults fed controlled diets of lower (control) and higher potassium as potato, French fries (FF), or K gluconate (Supp). Different superscript letters denote significant differences (*p* < 0.05) by ANOVA.

**Figure 3 nutrients-13-04399-f003:**
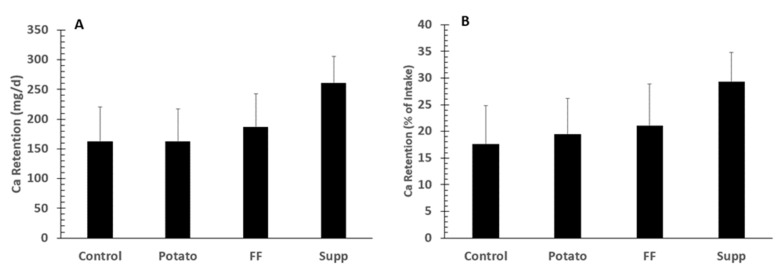
Means ±SE of calcium retention (*n* = 20) as absolute calcium (Ca) retention (**A**) or percent (%) of intake (**B**) in adults fed controlled diets of lower (control) and higher potassium as potato, French fries (FF), or K gluconate (Supp). There were no significant group differences among groups by ANOVA.

**Table 1 nutrients-13-04399-t001:** Baseline Characteristics of the Study Subjects (*n* = 30).

	All Mean (SD)	Male Mean (SD)	Female Mean (SD)
*n*	30	15	15
Age (years)	48.2 (15.0)	43.8 (13.7)	52.7 (15.4)
Height (cm)	172.2 (10.2)	179.4 (7.1)	165.0 (7.2)
Weight (kg)	93.86 (22.9)	99.1 (20.7)	88.6 (24.4)
BMI (kg/m^2^)	31.4 (6.1)	30.5 (4.8)	32.3 (7.2)

## Data Availability

All data are contained in the article or available upon reasonable request.

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
