# Peer review of "Short-Term Supplemental Dietary Potassium from Potato and Potassium Gluconate: Effect on Calcium Retention and Urinary pH in Pre-Hypertensive-to-Hypertensive Adults"

_nutrients, 2021, doi:10.3390/nu13124399_

Round 1
Reviewer 1 Report
no comments
Author Response
Please see the revised manuscript as attached PDF

Reviewer 2 Report
The Authors should clarify the main differences between the current paper and their previous paper "Short-Term RCT of Increased Dietary Potassium from Potato or Potassium Gluconate: Effect on Blood Pressure, Microcirculation, and Potassium and Sodium Retention in Pre-Hypertensive-to-Hypertensive Adults" Michael S Stone , Berdine R Martin , Connie M Weaver - Nutrients 2021 May 11;13(5):1610.
Author Response
Response to Reviewer 2- We clarified the purpose of this paper in the abstract and introduction and clearly distinguished these outcomes from the previous study as follows:
Abstract: In this secondary analysis of a study designed primarily to determine blood pressure effects,, we assessed the effects of potassium intake from potato sources and a potassium supplement on urinary Ca, urine pH, and Ca balance.
We also reduced redundancy in this manuscript with our previous paper (s).
Introduction-The purpose of this paper is to report findings of the secondary outcomes of a randomized, controlled feeding trial designed primarily to examine the effect of increased dietary potassium from potato sources and a potassium-gluconate supplement compared to a control diet on blood pressure. The primary outcomes have been previously reported [29].
This manuscript is a resubmission of an earlier submission. The following is a list of the peer review reports and author responses from that submission.
Round 1
Reviewer 1 Report
Since this is an interventional study, the Authors must provide a registration number (e.g. clinicaltrial.gov).
Bar graphs with error bars do not allow direct evaluation of the distribution of the data. The authors should present their data in scatter/dot plots (especially in case of a limited number of observations), showing the individual data points together with the average/error bars.
The analysis needs to be adjusted for potential confounders.
Relevant clinical parameters (including blood pressure values) must be included in Table 1 and in the subsequent analyses.
The manuscript is mainly descriptive and focused on its (not fully supported) conclusions, not adequately acknowledging the limitations of the study. The strengths and limitations of the study should be deeply addressed, taking into account sources of potential bias or imprecision: Discuss both direction and magnitude of any potential bias.
It is advisable to the Authors to incorporate a pictorial or cartoon representation of the main results of the study to increase the overall impact of the manuscript.
"eligble"
Author Response
Since this is an interventional study, the Authors must provide a registration number (e.g. clinicaltrial.gov).
This information was provided in the abstract (line 23) and has now also been added to the methods (lines 84-85).
Bar graphs with error bars do not allow direct evaluation of the distribution of the data. The authors should present their data in scatter/dot plots (especially in case of a limited number of observations), showing the individual data points together with the average/error bars.
Thank you for your suggestion. We added a figure (figure 4) to better show distribution of absolute Ca retention data.
The analysis needs to be adjusted for potential confounders.
Thank you for the suggestion. We had no cofounders that met adequate criteria to be included in any of our analyses. The crossover design eliminates the need for any confounder related to the participants’ characteristics.
Relevant clinical parameters (including blood pressure values) must be included in Table 1 and in the subsequent analyses.
Thank you for the suggestion, but since this study was not focused on blood pressure outcomes, and since the blood pressure data from this study have been published elsewhere, we did not include these data here. Baseline values (for reference) are included on lines 152-153. The aims of our study are now more clearly stated and contrasted with the previous publication on the primary outcomes.
The manuscript is mainly descriptive and focused on its (not fully supported) conclusions, not adequately acknowledging the limitations of the study. The strengths and limitations of the study should be deeply addressed, taking into account sources of potential bias or imprecision: Discuss both direction and magnitude of any potential bias.
Our main conclusions were that Ca retention did not differ among treatments and that further research is needed. It is difficult to understand how these conclusions were not supported by the results. The strengths and limitations of the study were expanded in lines 291-302:
Strengths of our study include the highly controlled diet, the use of a crossover design, and comparing food and supplement forms of potassium. The comparison of forms allowed insights into potential roles of the effect of bioactives or other constituents in potatoes. Duration of the study was adequate to achieve steady state calcium retention. A limitation of this study is that only one, relatively moderate Ca intake was included. A dose response design would have been stronger. It is possible that effects of potassium intake on Ca balance may have differing effects based on whether Ca intake is low, adequate, or high. As with all balance studies, there are potential sources of error with complete collection in only one direction, i.e. incomplete collection of urine and feces. Compliance with the diet was based on an honor system except for meals served in the clinic. Calcium absorption decreases with age [45], which may have affected intersubject variation, but not intrasubject variation in this crossover design. Our findings agree in part with the previous literature with Ca retention trending higher for the K-gluconate intervention vs. the control. However, we were most likely underpowered (~N = 50, based on post-hoc analysis).
It is advisable to the Authors to incorporate a pictorial or cartoon representation of the main results of the study to increase the overall impact of the manuscript.
Thank you for the suggestion, but we do not feel a pictorial representation of this study’s findings would strength the impact, which we have addressed at length in the discussion.
Reviewer 2 Report
“References” are not formatted as per journal requirements |
Lack of „Author Contributions” |
Lack of „Funding” |
Lack of „Institutional Review Board Statement” - despite the information that the details of the procedures can be found in another article, such consent should be clearly presented to readers in this article as well |
Line 48 - no explanation of what the DASH diet is - not everyone knows |
Line 49 - no explanation of what CTX is |
Line 73 and then 150 - not explained what SBP and DBP are |
Line 154 - 25 subjects completed all interventions, but that is not clear in Figure 1 |
Lines 154-155 - for clarity, the total number of people after the first sentence should be given in brackets (n=30) |
Figure 1 – the title should contain only the first sentence. Looking at the figure, it is not known what to do with the information that "one subject completed three, and four subjects completed two" - should the reader add something to something (and to what)? The whole diagram is not very clear - which stage does the "drop out" concern? What do the numbers of people in "control", "potato", "french fries" and "k salt" mean - nowhere else 30? |
Line 162 – there is no Figure 4.2 |
Lines 162-165 - Why the amount of potassium was lower than assumed |
Figure 2 – what the authors think about the relatively low of the urinary pH on the control diet, despite the lower sodium content (the role of sodium?) - sometimes it may indicate a potassium deficiency - how to explain this? |
Lines 273-287 - the authors discuss various forms of potassium in supplements, but not K-gluconate, the form they used themselves is missing - are there any studies on its bioavailability, if not, then its choice should be justified |
Lines 248-251 – this phrase suggests that content of potassium was ~300 mg/d lower in each „potato diet”, including FF (they are made from potatoes) – this is not true, as reflected by the comparable urinary Ca excretion after FF and supplement - please clarify this fragment |
Lines 289-299 - In my opinion, the limitation of this study is a wide age range, which may affect K absorption and Ca retention, which change with age and bone condition, and should be clearly mentioned |

Author Response
“References” are not formatted as per journal requirements
References have been formatted accordingly.
Lack of „Author Contributions”
Lack of „Funding”
Lack of „Institutional Review Board Statement” - despite the information that the details of the procedures can be found in another article, such consent should be clearly presented to readers in this article as well
Thank you, this information has now been added at the end of the manuscript (lines 315-331).
Line 48 - no explanation of what the DASH diet is - not everyone knows
DASH has now been defined (lines 48-49).
Line 49 - no explanation of what CTX is
CTX has now been defined (line 49).
Line 73 and then 150 - not explained what SBP and DBP are
SBP and DBP have now been defined (lines 74, 152).
Line 154 - 25 subjects completed all interventions, but that is not clear in Figure 1
Figure 1 has been adjusted to better represent this.
Lines 154-155 - for clarity, the total number of people after the first sentence should be given in brackets (n=30)
This has been added to the table title.
Figure 1 – the title should contain only the first sentence. Looking at the figure, it is not known what to do with the information that "one subject completed three, and four subjects completed two" - should the reader add something to something (and to what)? The whole diagram is not very clear - which stage does the "drop out" concern? What do the numbers of people in "control", "potato", "french fries" and "k salt" mean - nowhere else 30?
Figure 1, and Figure 1 legend have been adjusted for clarity.
Line 162 – there is no Figure 4.2
This reference has been deleted (line 164).
Lines 162-165 - Why the amount of potassium was lower than assumed
An explanation for this can be found in the discussion (lines 253-264).
Figure 2 – what the authors think about the relatively low of the urinary pH on the control diet, despite the lower sodium content (the role of sodium?) - sometimes it may indicate a potassium deficiency - how to explain this?
Thank you for your question. This would align with our study findings, as the control diet was designed to have an inadequate/lower, but typical, amount of dietary potassium (e.g., lower potassium = potentially lower urinary pH). This was misarranged in the results section, which may have been confusing, and has now been corrected. This has also been made clearer in the discussion (lines 211 -212).
Lines 273-287 - the authors discuss various forms of potassium in supplements, but not K-gluconate, the form they used themselves is missing - are there any studies on its bioavailability, if not, then its choice should be justified.
Thank you for your question, to our knowledge potassium gluconate has not been used in previous research to evaluate supplementation on calcium economy. However, potassium gluconate has been found to be highly bioavailable in previous research (doi: 10.3945/ajcn.115.127225), giving efficacy for its use in this study.
Lines 248-251 – this phrase suggests that content of potassium was ~300 mg/d lower in each „potato diet”, including FF (they are made from potatoes) – this is not true, as reflected by the comparable urinary Ca excretion after FF and supplement - please clarify this fragment
The potassium content in each of our potato food group phases (baked/boiled and French fries) was approximately 300mg/d lower than the supplement diet, which we confirmed with chemical analyses of all study diets. Numerous other balance studies have reported that potassium consumption/absorption and urinary excretion are not a 1:1 ratio (reviewed in reference 28). The difference in urinary excretion would imply that potassium supplementation may differentially effect Ca retention and/or bone health, as discussed throughout the discussion.
Lines 289-299 - In my opinion, the limitation of this study is a wide age range, which may affect K absorption and Ca retention, which change with age and bone condition, and should be clearly mentioned
Thank you for your suggestion. Potassium absorption is passive and does vary with age (doi: 10.3390/nu13051610), we have added the potential limitation of calcium absorption with age (lines 297-298), but it is unlikely to affect the response to intervention since each subject was their own comparator in this crossover design. One would expect absorption to change with age across all interventions similarly.
Reviewer 3 Report
The paper is about Short-term supplemental dietary potassium from potato and potassium gluconate: effect on calcium retention and urinary pH in pre-hypertensive-to-hypertensive adults. Considering the highest level of Nutrients, the paper needs serious improvements in order to be considered for publication. Information must be supplemented accordingly, the shape of the manuscript must be carefully polished/finished. Please see bellow my suggestions:
The worst - the number of patients involved in the study is much too low. To be relevant, a group in investigation must have at least 33 patients.
L82-84 the number/date of approval must be provided.
As the Instructions for authors request, please check and apply in the entire manuscript the rules for Acronyms/Abbreviations/Initialisms which should be defined at their first time they appear in each of three sections: the abstract; the main text; the first figure or table. When defined for the first time, the acronym/abbreviation/initialism should be added in parentheses after the written-out form.
Please replace y with year, d with day through the manuscript.
Figure 1 is blurred. Please replace it with a better quality one.
The groups of patients must be more detailed in an additional table. What risks factors for hypertension did they have? Were they smokers? Did they have family history of hypertension? Please discuss the impact of the prescribed diet on the values of blood pressure. The title mention prehypertensive and hypertensive adults but the results do not analyse the impact of rich potasium diet on values of blood pressure.
The Discussions are very limited. Please:
- detail the impact of calcium reduced excretion on blood pressure.
- make a summarising table with the beneficial and potential adverse effects of high potassium diet on blood pressure values and on other parameters.
- discuss if any other authors obtained similar results with young hypertensive patients.
- describe the impact of potassium supplementation in certain risk factors groups - it is known that hypertensive patients have also risk for atherosclerosis.
- better describe psychopathological correlation between potassium levels, sodium and calcium and other possible elements.
For all all above mentioned requests, you will find helpful and you may refer to Babes et al. Value of Hematological and Coagulation Parameters as Prognostic Factors in Acute Coronary Syndromes.Diagnostics. 2021; 11(5):850. https://doi.org/10.3390/diagnostics11050850 ; Stoicescu M.,et al. The role of increased plasmatic renin level in the pathogenesis of arterial hypertension in young adults. Rom. J. Morphol. Embriol., 52(1 Suppl.), 2011, 419-423. ; and Moisi, M.I.; et al. Framing Cause-Effect Relationship of Acute Coronary Syndrome in Patients with Chronic Kidney Disease. Diagnostics 2021, 11, 1518. https://doi.org/10.3390/diagnostics11081518. Illustrative figures in the discussion part would be relevant.
Additionally, the clinical utility in practice of your results should be better discussed/detailed. At the final of Discussion section please insert a paragraph regarding the strengths and weakness of your study.
References. Please check the Instructions for authors at the link https://www.mdpi.com/journal/nutrients/instructions and provide all data requested for each reference, writing them in the MDPI style.
Author Response
The paper is about Short-term supplemental dietary potassium from potato and potassium gluconate: effect on calcium retention and urinary pH in pre-hypertensive-to-hypertensive adults. Considering the highest level of Nutrients, the paper needs serious improvements in order to be considered for publication. Information must be supplemented accordingly, the shape of the manuscript must be carefully polished/finished. Please see bellow my suggestions:
The worst - the number of patients involved in the study is much too low. To be relevant, a group in investigation must have at least 33 patients.
Thank you for your comment. It is unclear where the rationale for “33 patients” comes from. The majority of the retention studies that were reviewed for the most recent Dietary Reference Intakes report on potassium and sodium (Reference #28) had fewer than 33 subjects. We also acknowledge in our limitations that we were most likely underpowered in this study (line 296). However, these data which are difficult and costly to obtain can be pooled with other studies in the future to strengthen the power.
L82-84 the number/date of approval must be provided.
This information has been added (lines 85-86).
As the Instructions for authors request, please check and apply in the entire manuscript the rules for Acronyms/Abbreviations/Initialisms which should be defined at their first time they appear in each of three sections: the abstract; the main text; the first figure or table. When defined for the first time, the acronym/abbreviation/initialism should be added in parentheses after the written-out form.
This has been adjusted through out the manuscript.
Please replace y with year, d with day through the manuscript.
This has been adjusted through out the manuscript.
Figure 1 is blurred. Please replace it with a better quality one.
Figure 1 has been replaced with a clearer version.
The groups of patients must be more detailed in an additional table. What risks factors for hypertension did they have? Were they smokers? Did they have family history of hypertension? Please discuss the impact of the prescribed diet on the values of blood pressure. The title mention prehypertensive and hypertensive adults but the results do not analyze the impact of rich potassium diet on values of blood pressure.
We have included all relevant participant characteristics in Table 1, as well as all inclusion/exclusion criteria (lines 73-79). Findings related to blood pressure have been previously published and do not relate to the primary objectives of this study (lines 66-70). Smoking was an exclusion criterion. The cross-over design of the study also mitigates any individual characteristics having any effect on the results.
The Discussions are very limited. Please:
- detail the impact of calcium reduced excretion on blood pressure.
- make a summarizing table with the beneficial and potential adverse effects of high potassium diet on blood pressure values and on other parameters.
- discuss if any other authors obtained similar results with young hypertensive patients.
- describe the impact of potassium supplementation in certain risk factors groups - it is known that hypertensive patients have also risk for atherosclerosis.
- better describe psychopathological correlation between potassium levels, sodium and calcium and other possible elements.
For all above mentioned requests, you will find helpful and you may refer to Babes et al. Value of Hematological and Coagulation Parameters as Prognostic Factors in Acute Coronary Syndromes. Diagnostics. 2021; 11(5):850. https://doi.org/10.3390/diagnostics11050850 ; Stoicescu M., et al. The role of increased plasmatic renin level in the pathogenesis of arterial hypertension in young adults. Rom. J. Morphol. Embriol., 52(1 Suppl.), 2011, 419-423. ; and Moisi, M.I.; et al. Framing Cause-Effect Relationship of Acute Coronary Syndrome in Patients with Chronic Kidney Disease. Diagnostics 2021, 11, 1518. https://doi.org/10.3390/diagnostics11081518. Illustrative figures in the discussion part would be relevant.
While we thank you for these thoughtful points, none of them appear to align with the primary objectives or hypothesis of the study which has been expanded as: “The purpose of this paper is to report findings of the secondary outcomes of a controlled feeding study designed primarily to examine the effect of increased dietary potassium from potato sources and a potassium-gluconate supplement compared to a control diet on blood pressure. The primary outcomes have been previously reported [29]. The goal of this research was to utilize a controlled feeding study to examine the effect of increased dietary potassium from potato sources and a potassium-gluconate supplement on the Ca economy compared to a control diet. It was hypothesized that the interventions with an increased potassium intake would decrease Ca urinary excretion and increase calcium retention and urinary pH compared to the control.” (lines 66-70).
We feel the discussion focuses on how potassium intake relates to calcium economy (a surrogate of bone health), which again was the primary objective of the study.
The blood pressure outcomes of this study were published previously.
Additionally, the clinical utility in practice of your results should be better discussed/detailed. At the final of Discussion section please insert a paragraph regarding the strengths and weakness of your study.
The potential clinical utility (bone health outcomes) of increased potassium intake effects on calcium economy can be found throughout the discussion, specifically lines 231-251, 267-276, 277-292.
There is a paragraph addressing the strengths and weaknesses of the study (lines 291-302).
References. Please check the Instructions for authors at the link https://www.mdpi.com/journal/nutrients/instructions and provide all data requested for each reference, writing them in the MDPI style.
References have been adjusted to match journal style.
Round 2
Reviewer 1 Report
The Reviewers' concerns were not addressed in a satisfactory manner.
Reviewer 3 Report
The authors responded to almost all my requests. However, the number of the investigated subjects remained very low, NOT statistically significant (under 33) fact which makes this study irrelevant.